# Role of Dietary Methyl Sulfonyl Methane in Poultry

**DOI:** 10.3390/ani13030351

**Published:** 2023-01-19

**Authors:** Yoo-Bhin Kim, Kyung-Woo Lee

**Affiliations:** Department of Animal Science and Technology, Konkuk University, Gwangjin-gu, Seoul 05029, Republic of Korea

**Keywords:** methyl sulfonyl methane, anti-oxidant, poultry

## Abstract

**Simple Summary:**

Commercial poultry production is linked to a variety of stresses that arise from the environment, nutrition, and metabolism of the host. These stressors may result in metabolic disorders and oxidative stress. It is pertinent to recognize and prevent signs of stress to minimize production loss and improve the health and well-being of chickens. The use of in-feed anti-oxidants to eliminate oxidative stress is a successful method of preventing tissue damage and augmenting the performance, health, and well-being of chickens. The burgeoning demand for in-feed anti-oxidants has spurred a review of the role of methyl sulfonyl methane as a dietary anti-oxidant in poultry.

**Abstract:**

Oxidative stress is defined as an imbalance between pro-oxidants and anti-oxidants within biological systems, leading to tissue damage and compromising the health of afflicted animals. The incorporation of dietary anti-oxidants into chicken diets has been a common practice to improve the performance, health, and welfare of the host by protecting against oxidative stress-induced damage. Methyl sulfonyl methane (MSM), a naturally occurring organosulfur compound found in various plant sources, has demonstrated various beneficial biological properties, including anti-inflammatory and anti-oxidant properties in both in vitro and in vivo studies. MSM has been utilized as a dietary supplement for humans for its anti-oxidant, analgesic, and anti-inflammatory properties. It has also been administered to domestic animals, including cattle, pigs, and chickens, owing to its recognized anti-oxidant effect. This review summarizes the biological and physiological functions of dietary MSM in poultry.

## 1. Introduction

Poultry production is an expanding animal industry that has provided an increasing amount of animal protein to address global food security [1,2]. However, high-efficiency poultry production has been hindered by various challenges, including disease, stress, animal welfare, and antibiotic ban, which might be considered a limiting factor in sustainable poultry production on a global scale. Avian diseases, for instance, result in considerable economic losses due to the mortality and morbidity of the afflicted flocks and are yet to be fully controlled [3]. Stress, an inevitable negative stimulus, could compromise the productive and reproductive performance of chickens [4,5]. Commercial poultry production is associated with a range of stresses originating from the environment, nutrition, and the metabolism of the host, which could lead to metabolic disorders and oxidative stress [6]. It is crucial to identify and prevent signs of stress to minimize production loss and improve the health and welfare of chickens. Dietary anti-oxidants have been used as a nutritional strategy for controlling oxidative stress in chickens [7]. Currently, various anti-oxidants, such as selenium, vitamins A, C, and E, and plant extracts, are used individually or in combination to prevent oxidative stress in poultry [1]. Many natural and synthetic anti-oxidants decrease or delay undesirable oxidative stress [8]. In light of the growing demand for in-feed anti-oxidants, this review explores the role of methyl sulfonyl methane (MSM) as a dietary anti-oxidant in poultry. MSM has gained potential as a functional feed additive in the animal feed industry as it is marketed as an anti-oxidant, anti-inflammatory, immune-modulating, and joint pain-relieving agent for human consumption. While the working mechanisms of dietary anti-oxidants and the physiological aspects and consequences of oxidative stress were well explored previously [1,6,9], here, we briefly touch upon oxidative stress and anti-oxidants in poultry, followed by the application of MSM in poultry.

## 2. Oxidative Stress as a Factor Affecting Poultry Production

Oxidative damage occurs in living animals as a result of an imbalance between the production of reactive oxygen or nitrogen species and the defense mechanisms of the animal against oxidative stress [10]. One of the detrimental effects of oxidative stress is molecular damage induction and the disruption of regular functions, specifically related to nucleic acids, lipids, and proteins [11], the cascade of which leads to the pathogenesis of diseases [12]. Oxidation is typically initiated by reactive oxygen species (ROS) produced via cellular metabolism [13]. ROS could have both deleterious and beneficial effects on living systems [11]. Low ROS levels interact with specific targets and play important roles in redox signaling, which is responsible for stress adaptation, homeostasis, and health maintenance [14]. However, high ROS exposure affects non-specific targets and causes oxidative distress, such as lipid peroxidation, DNA damage, or apoptosis, leading to compromised immunity, decreased resistance to various diseases, and decreased productive and reproductive performance of poultry [14,15]. Oxidative stress decreases anti-oxidant systems by reducing anti-oxidant enzyme activity. For instance, an oxidative stress-induced increase in free radicals lowers glutathione depletion, thereby causing a reduction in anti-oxidant enzyme levels [16]. It was well established that oxidative stress is a leading cause of the detrimental consequences of stress in poultry [5].

In poultry, oxidative stress is an important biological factor that affects growth and development [17]. Oxidative stress is a common problem encountered in commercial poultry production. Domestication and genetic selection for rapid growth, improved feed conversion, and high egg production rates have rendered domestic birds, including broilers, layers, and turkeys, particularly susceptible to oxidative stress [18]. Hot climates and oxidized feed are the most notable sources of oxidative stress in birds, leading to biological damage, serious health disorders, diminished growth rates, and, ultimately, economic losses [13]. The consumption of oxidized oil in the diet is considered a major contributor to oxidative stress in poultry [19]. Hence, utilizing oxidized oil in broiler feed may result in decreased shelf life and quality consistency of poultry meat [20]. Moreover, oxidative stress causes gastrointestinal disturbances, leading to poor gut health and subsequent production losses [18]. Oxidative stress negatively affects egg quality [21].

Poultry in intensive farming is frequently exposed to oxidative stress, which may damage body lipids, proteins, DNA, and other cellular constituents [4]. Both lipid and protein oxidation induced by oxidative damage are major threats to the quality of poultry meat and processed poultry products and leading causes of health concerns, consumer rejection, and economic losses [13]. Lipid oxidation is defined as a free radical chain reaction that consists of three steps: initiation, propagation, and termination [22,23]. Lipid oxidation causes the loss of nutritional and sensory values, as well as the formation of potentially toxic compounds that compromise meat quality and reduce shelf life [24]. Some oxidation products of lipids, including malondialdehyde and other dicarbonyl compounds, are toxic chemicals [25]. Poultry meat is especially prone to oxidative deterioration due to its high content of polyunsaturated fatty acids, and the oxidation of lipids is one of the most detrimental factors causing quality loss in animal foods [8]. Protein oxidation is commonly linked to the loss of nutritional value and a decrease in muscle protein functionality, leading to increased water loss, weak protein gel formation, and less stable emulsions [26]. The oxidative modification of proteins and the formation of carbonyl derivatives have harmful effects on meat quality, and protein carbonyl content increases during storage [27]. Proteins are major targets of ROS; the accumulation of oxidized products in muscle tissue leads to meat quality deterioration [20].

## 3. Synthetic and Natural Anti-Oxidants

Anti-oxidants inhibit the oxidation of other molecules [17], ameliorate oxidative tissue damage [28], and scavenge reactive oxygen and nitrogen species to reduce oxidative stress. The most important step in balancing oxidative damage and anti-oxidant defense in the animal body is to enhance anti-oxidant capacity by optimizing the dietary intake of anti-oxidants [6]. The oxidation process may be retarded by the use of dietary anti-oxidants, which play a critical role in protecting cellular components from potentially damaging ROS, thereby maintaining homeostasis and optimal cellular functions [29].

Many natural and synthetic anti-oxidants decrease or delay undesirable oxidative processes [8]. Based on their source, anti-oxidants may be classified as synthetic or natural. Synthetic anti-oxidants have been widely used as food preservatives because of their effectiveness and relatively low cost [30]. Natural anti-oxidants, found in plant parts, such as leaves, bark, seeds, and fruits, include tocopherols, vitamin C, and flavonoids, whereas synthetic anti-oxidants include butylated hydroxytoluene (BHT), butylated hydroxyanisole (BHA), and ethoxyquin [31]. However, the use of synthetic anti-oxidants has fallen under scrutiny due to their potential toxicological effects on both humans and animals [32,33]. It is well acknowledged that natural anti-oxidants play important roles in maintaining the health, welfare, and productive and reproductive performance of chickens [14]. Therefore, their addition to chicken diets has been commonly practiced to increase the internal concentration of anti-oxidants that slows down the oxidative effects in meat [6,33].

Many natural anti-oxidants are provided in chicken diets, while a range of other anti-oxidant compounds are synthesized in the body (e.g., glutathione or anti-oxidant enzymes), and a close balance between anti-oxidants and pro-oxidants in the cell, digestive tract, and the whole body is responsible for the maintenance of chicken health and productive and reproductive performance [34]. Indeed, natural dietary anti-oxidants improve the growth performance, carcass characteristics, and fatty acid profiles of broiler chicken meat [35]. In addition, dietary anti-oxidants improve egg quality in laying hens [36] and retard lipid oxidation in eggs [37]. During our attempts to identify the sources of natural anti-oxidants, MSM has gained attention owing to its wide range of biological properties, including free radical scavenging properties [38,39].

## 4. Methyl Sulfonyl Methane

MSM (molecular formula: (CH_3_)_2_SO_2_) is a stable oxidized metabolite of dimethyl sulfoxide (DMSO) containing 34% sulfur on a weight basis [40,41]. MSM is a small sulfur-based molecule comprising a sulfur atom with two double-bonded oxygen atoms and two methyl groups (Figure 1). It is a white crystalline powder easily soluble in water and utilized as complementary and alternative medicine. It has a variety of names, including dimethyl sulfone, methyl sulfone, sulfonylbismethane, organic sulfur, and crystalline dimethyl sulfoxide [38]. MSM exists naturally in various foods, such as milk, fruits, vegetables, grain crops, and animal tissues [42]. Cow’s milk is the richest source of MSM, containing approximately 3.3 ppm [42]. Other foods containing MSM include coffee (1.6 ppm), tomatoes (trace to 0.86 ppm), tea (0.3 ppm), Swiss chard (0.05–0.18 ppm), beer (0.18 ppm), corn (up to 0.11 ppm), and alfalfa (0.07 ppm) [42]. MSM is naturally present in chickens organism. MSM is found in the plasma (71.4 μg/mL), liver (459.5 μg/g), spleen (186.2 μg/g), heart (137.6 μg/g), kidney (131.5 μg/g), brain (124.9 μg/g), cecal tonsil (102.2 μg/g), hock joint (177.5 μg/g), and abdominal skin (143.0 μg/g) of non-MSM-fed broiler chickens [43]. Rasheed et al. [43] reported that MSM could be naturally synthesized in chickens fed an MSM-free diet. MSM was also detected at a concentration of 0.6 g/kg of egg albumen in laying hens [44]. However, Kim et al. [45] reported that the basal diet of laying hens already contained 0.31 g MSM/kg. Thus, the natural presence of MSM in chicken products might be endogenous, exogenous, or both.

MSM could be synthesized naturally using algae or marine microorganisms or chemically [46]. Synthetically produced MSM is manufactured via the oxidation of DMSO with hydrogen peroxide, followed by purification via either crystallization or distillation [38]. There are no differences in the biological activities, structure, and safety between naturally occurring and chemically synthesized MSM [38]. Owing to the low concentration of MSM in nature, synthetically produced MSM has been widely used as a source of MSM in various foods and health supplements for human consumption.

Although MSM is believed to be non-toxic [47], its safety was evaluated in several rodent studies [48]. MSM is considered safe at the recommended oral dosages for humans [49]. Magnuson et al. [48] observed no evidence of toxicity after MSM ingestion in pregnant rats. Indeed, the US Food and Drug Administration (FDA) designation is classified as “generally regarded as safe (GRAS)”, which allows MSM to be added to foods and dietary supplements without direct regulation. Under the FDA GRAS notification, MSM is considered safe at dosages below 4845.6 mg/day [50]. Upon ingestion, MSM is rapidly absorbed in the intestine via passive diffusion, well distributed throughout the body, and rapidly excreted primarily through the urine [48,51]. Collectively, there is a growing body of evidence of safety in the use of MSM as a dietary supplement, as it is predominantly eliminated.

Once absorbed, dietary MSM may be metabolized to produce metabolites with potential medical value, such as sulfur-containing amino acids [52]. MSM has an enhanced ability to penetrate cell membranes inside the body and cross the blood-brain barrier [38,53]. Recent in vivo studies with radiolabeled MSM have suggested that this compound is rapidly metabolized in tissues [54] because of its high aqueous solubility [51]. Thus, MSM may exhibit physiological activities at the cellular and tissue levels [54]. Earlier studies in rodents demonstrated that sulfur from MSM could be incorporated into tissue proteins [55]. Additionally, most MSM metabolites are excreted via the urine [52]. Thus, dietary MSM is required to maintain constant biological levels for beneficial effects due to its rapid metabolism and excretion [54]. In addition, Kim et al. [44] reported that dietary MSM in laying hen diets could be effectively and linearly transferred into egg albumens (Figure 2). According to Kim et al. [44], MSM was detected at 178.9 mg/egg in a laying hen fed 4.0 g MSM/kg diet, and this MSM-enhanced egg provides an adult with 4% of the recommended intake.

Sulfur has an atomic weight of 32.064 with an atomic number of 16 and is represented by the chemical symbol “S” [56]. Sulfur represents ~0.3% of the total body mass, is the 7th most abundant element in the body [51], and is incorporated into amino acids, proteins, enzymes, and micronutrients [57]. Thus, sulfur is an important element for animal growth required for the formation of many S-containing compounds in host cells [58]. MSM is a sulfur donor for macromolecules, including methionine, cysteine, homocysteine, and taurine [38]. Sulfur-containing amino acids influence the cellular redox state and the ability to detoxify free radicals, ROS, and toxic substances, thus substantially contributing to the maintenance and integrity of cellular systems [59]. Although limited, there is evidence that sulfur in MSM may be incorporated into sulfur-containing amino acids (e.g., methionine and cysteine) upon ingestion in animals [55]. However, it is unlikely that dietary MSM is directly incorporated into endogenous methionine and cysteine in chickens, as chickens are unable to synthesize endogenous sulfur-containing amino acids from MSM-derived sulfur. Wong et al. [51] proposed that rather than being directly absorbed, MSM would first be incorporated into gut bacteria, and then biologically available sulfur-containing amino acids would be produced through bacterial assimilation, as observed in mice studies. 

MSM is a popular dietary supplement in over-the-counter sales [43] and is often combined with glucosamine as a natural alternative pain reliever [60]. In addition, MSM acts to relieve inflammation, joint and muscle pain, and oxidative stress and exerts anti-oxidant capacity. MSM is also known for its effect on the modulation of oxidative stress and anti-oxidant defense [61]. While it does not chemically neutralize ROS in stimulated neutrophils, it does suppress the generation of superoxide, hydrogen peroxide, and hypochlorous acid inside the cells [62]. Kim et al. [63] reported that MSM inhibits inflammation by acting on the inflammatory mediators, inducible nitric oxide synthase, cyclooxygenase 2, prostaglandin E2, IL-6, and tumor necrosis factor-alpha, by downregulating nuclear factor kappa B signaling. Therefore, MSM with anti-inflammatory and anti-oxidant capacities is expected to exhibit a beneficial balance between the immune response and inflammatory tissue damage. Butawan et al. [38,39] extensively summarized the working mechanism of MSM as a free radical scavenging anti-oxidant in vitro and in vivo.

## 5. Effects of Dietary MSM on Biomarkers of Oxidative Stress and Anti-Oxidative Capacity

Animal studies using MSM as the primary treatment for experimentally induced injuries show reductions in malondialdehyde (MDA) [61,64,65,66,67,68,69,70], glutathione disulfide (GSSG) [61], myeloperoxidase (MPO) [66,69,70], and protein carbonyl (PC) [61,68] and increases in glutathione (GSH), catalase (CAT), superoxide dismutase (SOD), total anti-oxidant capacity (TAC), and glutathione peroxidase (GPx) [16,19,44,45,61,64,65,66,67,68,69,70,71]. The exact mechanism by which MSM attenuates oxidative stress markers is not well established, and further exploration is needed [61]. The effects of MSM on the biomarkers of oxidative stress in poultry are summarized in Figure 3.

The thiobarbituric acid reactive substances (TBARS) assay measures carbonyl compounds derived from lipid peroxidation in the body [19]. The principle of the assay depends on the colorimetric determination of the pink pigment product resulting from the reaction of one molecule of MDA with two molecules of thiobarbituric acid [67]. MDA is one of the most commonly used indicators of lipid peroxidation in cells [66]. In addition, MDA is formed during oxidative injury to DNA, proteins, and carbohydrates [72]. The reduced MDA levels may imply that MSM exerts potent anti-oxidant activity to scavenge free radicals [65]. MSM reduced MDA levels when supplemented at 0.03% for 42 days in male ducklings [64], 0.2% for 84 days in laying hens [45], and 0.3% for 42 days in female ducklings [65]. Hwang et al. [64] observed that supplementing the diet with a combination of oriental herbal medicine residue and MSM resulted in a low TBARS concentration in the breast meat (musculus pectoralis) of ducks, indicating less lipid oxidation. After treatment with MSM, the MDA levels in both the liver and lung tissues decreased [66]. Kamel and El Morsy [67] reported that pre-treatment with MSM reduced oxidative damage. Nakhostin-Roohi et al. [61,68] reported that MDA levels were higher in the placebo group than those in the MSM group. Furthermore, MSM has direct radical-scavenging activity, which may play a role in lowering the serum levels of MDA. In contrast, broilers supplemented with MSM (0.05%) for 21 days showed no significant changes in MDA in the plasma or liver samples [19]. Rasheed et al. [19] reported that MDA levels were either unaffected at most time points or low in some cases.

Glutathione is a primary endogenous anti-oxidant [19]. GSH is the most important intracellular anti-oxidant thiol, and its main source is the liver. Being a source of sulfur, MSM provides organic sulfur for the synthesis of GSH. Therefore, it counteracts the depletion of GSH in cells [69,70]. The ability of MSM to prevent GSH depletion under other oxidative conditions, such as exercise or chemically induced oxidative stress, was reported [16,61,66,68,69,70]. Marãnon et al. [16] reported that supplementation with MSM induced an increase in GSH levels, as expected, since MSM metabolism provides one of the precursors needed for GSH synthesis, thereby counteracting GSH depletion. MSM treatment significantly increased GSH levels in colonic tissues [69]. An increase in colonic GSH content may explain some of the beneficial effects of MSM in experimental colitis [69]. Contrary to these findings, Rasheed et al. [19] reported that the GSH concentration did not increase in MSM-fed groups, suggesting that MSM may not be effective at elevating GSH levels in the body or that the MSM dosage used in the study was insufficient to elicit a positive response.

Myeloperoxidase (MPO), a peroxidase enzyme produced by neutrophils, serves as an indicator of the inflammatory response. Reduced MPO activity was associated with the anti-inflammatory effects of MSM [69]. Furthermore, MPO catalyzes the production of potent oxidants and leads to tissue injury during oxidative stress and inflammation [66]. Schwarz et al. [73] also identified MPO as a key oxygen-dependent enzyme in neutrophils, which, if released into the local tissue or systemic circulation, could induce oxidative stress and cytotoxicity. In intestinal inflammation processes, MPO activity is a marker of neutrophil infiltration in the colon. Amirshahrokhi et al. [69] observed that treatment with MSM significantly reduced MPO activity in colonic tissue samples from a colitis group. Moreover, pre-treatment with MSM led to the restoration of tissue MPO activity toward normal levels [66,70]. This suggests that the protective effect of MSM on the liver tissue could be due to its anti-inflammatory action, which coincides with other studies [66,70].

Three major anti-oxidant enzymes, namely SOD, GPX, and CAT, are responsible for the detoxification of radicals at their inception [5]. These enzymes are integral in the scavenging of oxidative radicals, the reduction of oxidative damage, and the maintenance of cell structure [19]. There are many different mechanisms through which anti-oxidants exert their protective effects against oxidative damage.

Superoxide dismutase (SOD) is an enzyme that catalyzes the dismutation of superoxide anions into oxygen and hydrogen peroxide. Studies have indicated that MSM-fed ducks had significantly higher serum SOD levels than the control ducks [64]. In addition, Yan et al. [65] reported that high serum SOD activity indicates that MSM may provide efficient free radical-scavenging activity in Pekin ducks. Other researchers have also reported a significant increase in SOD activity in laying hens [44].

Glutathione peroxidase (GPX) is a common enzymatic anti-oxidant in livestock [74]. GPX eliminates H_2_O_2_ using GSH as a substrate [75]. MSM-supplemented broilers had higher liver GPX activity at day 21 than non-MSM-supplemented broilers [19]. Similarly, serum GPX activities were higher in ducks in the MSM treatment group than those in the control group [65].

Catalase is an important peroxisomal anti-oxidant enzyme that catalyzes the decomposition of hydrogen peroxide into water and oxygen. MSM-fed ducks had significantly higher CAT levels than control ducks [64], which may have provided more efficient free radical scavenging activity.

TAC, another good indicator of redox potential, was not affected by MSM supplementation in the liver of broilers [19]. However, broilers supplemented with MSM exhibited a higher plasma TAC than non-MSM-supplemented broilers [19]. Additionally, supplementation with MSM increased TAC levels in the serum of ducks [65] and laying hens [44,45,71].

## 6. Effect of Dietary MSM on the Immune Response

Sulfur-containing compounds, including MSM, are essential in supporting the immune response [56]. MSM modulates the immune response by mediating the interplay between oxidative stress and inflammation [38]. The anti-inflammatory and anti-oxidant capacity of MSM may help maintain a beneficial balance between the immune response and inflammatory tissue damage [76]. The effects of MSM on the immune responses in poultry are summarized in Table 1.

Interleukins (IL-2 and IL-6) are involved in the immune response, stimulating the proliferation of activated natural killer cells, B lymphocytes, T lymphocytes, and antibodies, as well as the production of IgA, IgM, and IgG. The broad effects of IL-6 were implicated in the maintenance of chronic inflammation [78]. Yan et al. [65] reported that MSM supplementation at 0.3% increased serum IL-2 and IL-6 levels in ducks. In laying hens, the IL-2 concentration and the ratio of CD4+ and CD8+ in the blood were generally high at 0.4% MSM [77]. Furthermore, MSM supplementation (0.3%) decreased the levels of the pro-inflammatory cytokines TNF-α and IFN-γ, indicating the anti-inflammatory effect of MSM on ducks. Amirshahrokhi et al. [69] reported that MSM reduced the TNF-α levels of rats. However, Rasheed et al. [76] reported that the gene expression pattern of pro-inflammatory (IL-1b and IFN-g) and anti-inflammatory (IL-10) cytokines showed no evidence of the beneficial effects of MSM supplementation when fed to birds.

## 7. Effect of Dietary MSM on Liver Function

The liver serves multiple metabolic functions by regulating the glucose and lipid stores and producing secretions to degrade metabolic waste, drugs, and chemicals into excretable compounds [39]. Liver disease is an important cause of morbidity and mortality globally. Elevated levels of liver-specific enzymes may indicate decreased liver function [66]. MSM improves liver function and suppresses hepatic tumorigenesis by activating apoptosis [79]. MSM (0.15%)-treated birds exhibited decreased aspartate aminotransferase (AST), creatine phosphokinase (CPK), and glutamate dehydrogenase (GLDH) concentrations [43]. This observation was in congruence with similar studies conducted in laying hens, with the AST levels in the serum decreasing following the addition of MSM [77].

Similar effects were observed in rats and mice [66,67]. The treatment of rats with MSM (0.04%) reduced alanine aminotransferase (ALT) and AST activity by 46 and 23.5%, respectively [67]. Kim et al. [79] and Bohlooli et al. [70] observed an improvement in liver functions with low AST and ALT levels. There was a significant reduction in the plasma levels of ALT, gamma-glutamyl transferase (GGT), and alkaline phosphatase (ALP) in mice treated with MSM [66]. Some studies reported that MSM prevented liver toxicity, as confirmed by the marked decline in hepatic biomarkers.

Conversely, Jiao et al. [41] indicated that supplementation with MSM (0.2%) increased ALT levels, although the levels of AST did not change. However, the ratio of AST/ALT showed a considerable decline. According to Jiao et al. [41], the ratio of serum aspartate to alanine aminotransferase (AST/ALT) levels is often used as an indicator for identifying liver diseases. Therefore, a positive effect could protect the liver of broilers fed diets supplemented with MSM. There are also reports that the supplementation with MSM (0.03%) does not affect AST and ALT levels in ducks [64]. These results suggest that MSM may be effective in inhibiting hepatic tumor growth. A summary of the liver function studies is listed in Table 2.

## 8. Effect of Dietary MSM on Performance

Several studies have examined the effects of supplementation of poultry diets with MSM [19,43,64,65,78,80] and have not found any related adverse effects on bird growth performance. Rasheed et al. [19] included 0.05% MSM in the base diet and found no effect on growth performance parameters. Similar results were obtained in a previous study that involved 0.05% MSM oral gavage daily for 21 days [43]. Most previous studies on MSM [64,81] also reported no effects on growth performance, such as feed intake (FI), body weight gain (BWG), and feed conversion ratio (FCR). Cho et al. [82] found that dietary MSM treatment (0.01%) did not affect BWG, FI, or feed efficiency in growing-finishing pigs. Park et al. [80] did not observe any effect of MSM (0.1%) on laying performance (i.e., egg production rate, egg weight, and FCR). However, Jiao et al. [41] found that supplementation with MSM (0.2%) increased BWG and reduced FCR on days 1–29 in broilers. In addition, Yan et al. [65] showed that the inclusion of MSM (0.3%) improved final BW and BWG during days 22–42 and days 1–42, as well as reduced FCR during days 22–42. Lim et al. [77] observed an increase in egg production in laying hens fed 0.4% MSM in their diet.

Overall, MSM may have a direct effect on growth performance in poultry. The absence of negative effects on growth performance in previous experiments indicates that the birds exhibited normal feeding behavior and growth under high doses of MSM. The effects of MSM on the growth performance parameters of poultry are summarized in Table 3.

## 9. Effect of Dietary MSM on Meat Quality

Poultry meat quality is a complex concept influenced by various factors, including composition, nutrients, colorants, water-holding capacity (WHC), tenderness, functionality, flavors, spoilage, and contamination [83] and subject to consumer preference [84]. Table 4 summarizes the effects of MSM on meat quality in poultry.

WHC is a crucial factor for both raw and processed meat products, as a high WHC could impact sensory characteristics and meat functionality [85]. In Cherry Valley male ducks, supplementation with MSM resulted in a significantly lower percentage of moisture loss (PML) and higher WHC than those in the control group [64]. PML was determined as described by Shon and Chin [86]. Additionally, the inclusion of MSM increased the WHC in the breast meat of Pekin ducks [65].

Suryanti et al. [87] reported that duck meat containing high levels of unsaturated fats is susceptible to oxidation, which may lead to rancidity and the deterioration of flavor and color. In general, myoglobin pigments, responsible for the red color of raw meat, may be oxidized to metmyoglobin during storage, resulting in meat discoloration. Fernández-López et al. [88] reported that increased lightness and decreased redness are associated with the oxidation of myoglobin to metmyoglobin. Yin and Faustman [89] also reported that the rate of meat discoloration is related to the rate of myoglobin oxidation induced by lipid oxidation. Accordingly, the presence of anti-oxidant or bioactive compounds could delay the formation of metmyoglobin. Yan et al. [65] reported that increased redness (a*) may be due to the anti-oxidant effect of MSM, which delays metmyoglobin formation. Hwang et al. [64] also found increased redness in the breast meat of MSM-fed ducks. However, Jiao et al. [41] suggested that the increase in redness after MSM dietary supplementation is associated with negative effects on broiler meat quality.

It has been well established that lightness and pH in poultry meat are negatively correlated [90]. A pH of 5.9 is an indicator of rigor mortis [91]. In addition, pH has an important function in meat products because its decline rate in animals post-mortem is directly related to meat tenderness [92]. Hwang et al. [64] reported no differences in cooking loss or pH in the MSM-treated duck group, which varied between 5.72 and 5.80. Jiao et al. [41] also found no effect on the pH of broiler meat.

## 10. Conclusions

Domestic animals, including chickens, are exposed to free radicals as a part of their biological processes; therefore, an integrated anti-oxidant system is needed to prevent damage to biologically relevant molecules, including DNA, proteins, and lipids. Ameliorating oxidative stress using in-feed anti-oxidants is an important tool to prevent tissue damage and improve the performance, health, and welfare of chickens. MSM is a naturally occurring organosulfur compound with broad biological effects. From this extensive review, it could be concluded that dietary MSM improves the performance, liver function, meat quality, immune response, and anti-oxidant capacity of chickens. Thus, dietary MSM could be used as an anti-oxidant agent for chickens. Cost-benefit improvements may be expected following supplementation with MSM as a functional anti-oxidant in poultry. Finally, based on its antimicrobial properties [93], dietary MSM may be further explored as a gut health enhancer by affecting the gut microbiome and gut barrier integrity in chickens.

## Figures and Tables

**Figure 1 animals-13-00351-f001:**
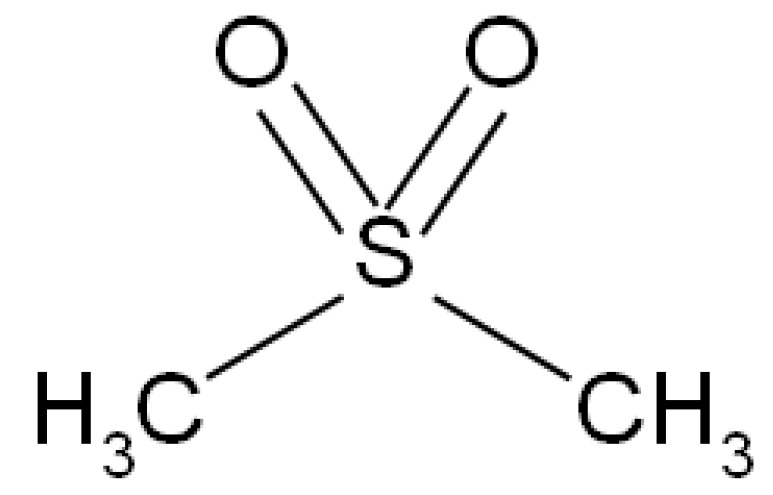
Structure of methyl sulfonyl methane.

**Figure 2 animals-13-00351-f002:**
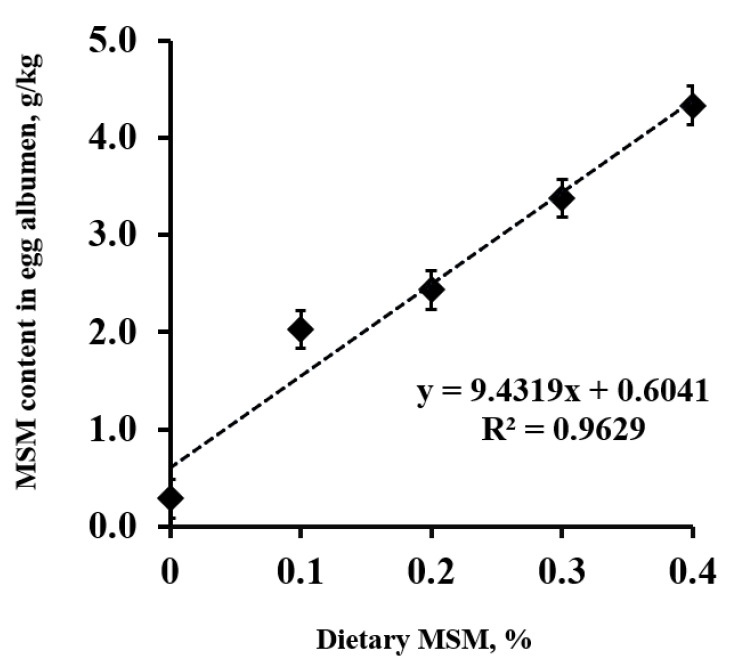
Methyl sulfonyl methane (MSM) content (g/kg) in lyophilized egg albumens from laying hens fed on a control diet or an MSM-enriched diet containing 1.0, 2.0, 3.0, and 4.0 g MSM per kg of diet. Source: adapted from Kim et al. [44].

**Figure 3 animals-13-00351-f003:**
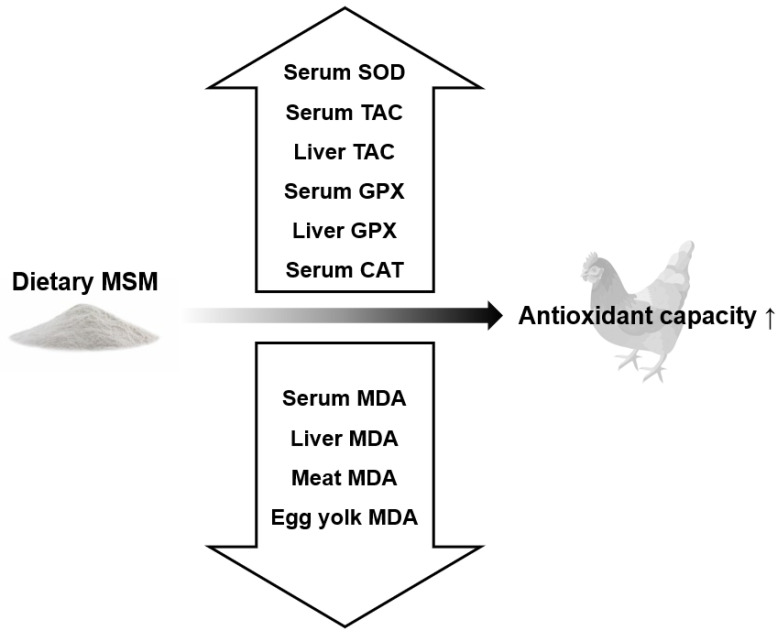
Effects of dietary methyl sulfonyl methane on the biomarkers of oxidative stress in poultry. SOD: superoxide dismutase; TAC: total anti-oxidant capacity; GPX: glutathione peroxidase; CAT: catalase; MDA: malondialdehyde.

**Table 1 animals-13-00351-t001:** MSM inclusion in the diet and its effects on the immune response in comparison with controls.

Species	Sample	Level of Inclusion	Results	Ref.
Pekin female ducklings	Serum	0.3%	No effects on IgG; increased IL-2 by 11.4%; increased IL-6 by 15.4%; decreased IFN-γ by 14.1%; decreased TNF-α by 12.8%	[65]
Lohmann brown laying hens	Serum	0.4%	Increased IL-2 by 58.2%	[77]

TNF-α: tumor necrosis factor-α; IFN-γ: interferon gamma; IgG: immunoglobulin G; IL-2: interleukin-2; IL-6: interleukin-6.

**Table 2 animals-13-00351-t002:** MSM inclusion in diets and the effect on liver function in comparison with controls used in different traits.

Species	Sample	Level of Inclusion	Results	Ref.
Ross 308 male broiler	Serum	0.2%	ALT increased by 22%; No effects on AST	[41]
Ross 308 male broiler	Serum	0.15%	AST decreased by 46.4%; CPK decreased by 48.7%; GLDH decreased by 54.9%	[43]
Cherry Valley male ducklings	Serum	0.03%	AST and ALT remained unchanged	[64]
Lohmann brown laying hens	Serum	0.4%	AST decreased by 10.6%	[77]
Adult male Swiss Wistar mice	Plasma	0.05%	ALT decreased by 35.5%; ALP decreased by 62%; GGT decreased by 303.8%	[66]
Female Sprague-Dawley rats	Serum	0.04%	ALT decreased by 46%; AST decreased by 23.5%	[67]
Transgenic male mice	Plasma	0.01%	ALT and AST decreased	[79]
Male Sprague-Dawley rats	Serum	0.01%	ALT and AST decreased	[70]

ALP: alkaline phosphatase; ALT: alanine aminotransferase; AST: aspartate aminotransferase; CPK: creatine phosphokinase; GGT: gamma-glutamyl transferase; GLDH: glutamate dehydrogenase.

**Table 3 animals-13-00351-t003:** MSM inclusion in diets and effect on growth performance traits in comparison with control used in different traits.

Species	House	Level of Inclusion	Experiment Period	Results	Ref.
Ross 308 male broiler	Battery cages	0.05%	1 to 21 D of age	No effects on BW and BWG	[19]
Ross 308 male broiler	Battery cages	0.15%	1 to 21 D of age	No effects on BWG, FI, and FCR	[43]
Ross 308 male broiler	Battery cages	0.2%	1 to 29 D of age	BWG increased by 2.8%, and FCR decreased by 2.6% compared to the control group for the whole period of the trial (days 1–29)	[41]
Cherry Valley male ducklings	Pens	0.03%	21 to 62 D of age	No effects on BWG, FI, and FCR	[64]
Pekin female ducklings	Battery cages	0.3%	1 to 42 D of age	No effects on BWG, FI, and FCR (days 1–21); BWG increased by 3.3% compared to the control group for the whole period of the trial (days 1–42)	[65]
Lohmann Brown laying hen	Battery cages	0.4%	31-wk-old (experiment lasted 24 weeks)	No effects on FI, FCR, and egg weight; egg production increased by 3.7% compared to the control group (weeks 17–24)	[77]
Lohmann Brown laying hen	Battery cages	0.1%	35-wk-old (experiment lasted 5 weeks)	No effects on egg production, egg weight, and FCR	[80]
Lohmann Brown-Lite laying hen	Battery cages	0.2%	73-wk-old (experiment lasted 12 weeks)	No effects on FI, FCR, and egg production	[71]
Lohmann Brown-Lite laying hen	Battery cages	0.1–0.4%	73-wk-old (experiment lasted 12 weeks)	No effects on FI, FCR, egg production, egg weight, and egg mass	[44]

BW: body weight; BWG: body weight gain; FCR: feed conversion ratio; FI: feed intake.

**Table 4 animals-13-00351-t004:** MSM inclusion in the diets and effects on meat quality in comparison with controls.

Species	Sample	Level of Inclusion	Collection Day (for the Feeding Period)	Results	Ref.
Ross 308 male broiler	Breast meat	0.2%	Day 29	No effect on pH; redness increased by 3.1%; drip loss decreased by 4.1%	[41]
Cherry Valley male ducklings	Breast meat	0.03%	Day 42	No effects on the cooking loss and pH; PML decreased by 15.7%; WHC increased by 5.3%; redness increased by 9.5%	[64]
Pekin female ducklings	Breast meat	0.3%	Day 42	pH increased by 2.5%; WHC increased by 10.4%; redness increased by 38.4%; drip loss decreased by 5.1%	[65]

PML: percentage moisture loss; WHC: water-holding capacity.

## Data Availability

All data generated during the study are included in the published article(s) cited within the text and acknowledged in the reference section.

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
