# Peer review of "Role of Dietary Methyl Sulfonyl Methane in Poultry"

_animals, 2023, doi:10.3390/ani13030351_

Round 1

Reviewer 1 Report

Dear Editor 

The aim of the article animals-2143060 ‘’Role of Dietary Methyl Sulfonyl Methane in Poultry’’ is to summarize the biological and physiological functions of dietary MSM for poultry.

This is a very interesting review study and deals with an interesting topic and fits with the scope of the journal. However, the manuscript needs a serious English language editing. 

I suggest the acceptance under a major revision based on the following major and minor comments. 

Comments for the authors

Major comments

-L139: you should be better to report studies and results by the use of methyl sulfonyl methane (MSM) and in other animal species, especially in monogastric

- L319-419: you could add a table, summarizing the results of previous studies based on the use of MSM in poultry

- you should include a paragraph focusing on the cost-benefit use of MSM in poultry 

- you should check carefully the manuscript for grammar mistakes

Minor comments

§  L7-8: .. of stress originating from the environment around and the nutrition and metabolism..

§  L10: .. of stress to minimize..

§  L11: .. in-feed antioxidants..

§  L13: .. demand for in-feed antioxidants..

§  L18: … prove the performance..

§  L22: .. s a dietary supplement for humans as an antioxidant, pain..

§  L38: .. of stress originating from the environment around and the nutrition..

§  L49: … n the animal feed industry as it is marketed as an antioxidant, anti-inflammatory..

§  L74: .. is a very common problem..

§  L81: .. is considered to be a major contributor to oxidative stress

§  L89: .. the quality of poultry meat and processed poultry products and are a leading cause of..

§  L99: .. leading to increased water losses..

§  L116: .. evidence of the oxidative process..

§  L125: .. maintaining the health, welfare

§  L148: .. richest source..

§  L158: .. hat the natural presence…

§  …….

Author Response

The aim of the article animals-2143060 ‘’Role of Dietary Methyl Sulfonyl Methane in Poultry’’ is to summarize the biological and physiological functions of dietary MSM for poultry.

This is a very interesting review study and deals with an interesting topic and fits with the scope of the journal. However, the manuscript needs a serious English language editing. 

I suggest the acceptance under a major revision based on the following major and minor comments. 

Authors’ Response: We have now English-edited the manuscript per the Reviewer’s recommendation.

Comments for the authors

Major comments

L139: you should be better to report studies and results by the use of methyl sulfonyl methane (MSM) and in other animal species, especially in monogastric

Authors’ Response: In Section 4, we have highlighted MSM per se introducing novel MSM in general. We have described the effects of dietary MSM on antioxidant capacity, immune response, liver function, performance, and meat quality in poultry. (Sections 5, 6, 7, 8, and 9).

L319-419: you could add a table, summarizing the results of previous studies based on the use of MSM in poultry

Authors’ Response: Per the Reviewer’s recommendation, we have now added four Tables that summarized the results of previous studies in Table 1, Table 2, Table 3, and Table 4.

You should include a paragraph focusing on the cost-benefit use of MSM in poultry 

Authors’ Response: We agree with the Reviewer’s comment that economic benefit use of MSM is the priority consideration in poultry nutrition. We have now indicated the cost-benefit use of MSM in poultry in the text as follows (Lines 419-420):

Cost-benefit improvements can be expected following supplementation with MSM as a functional antioxidant in poultry.

You should check carefully the manuscript for grammar mistakes

Authors’ Response: Per the Reviewer’s comment, we tried our best to improve the manuscript. We have checked the manuscript for grammar mistakes.

Minor comments

L7-8: of stress originating from the environment around and the nutrition and metabolism..

L10: of stress to minimize..

L11: in-feed antioxidants..

L13: demand for in-feed antioxidants..

L18: prove the performance..

L22: s a dietary supplement for humans as an antioxidant, pain..

L38: of stress originating from the environment around and the nutrition..

L49: n the animal feed industry as it is marketed as an antioxidant, anti-inflammatory..

L74: is a very common problem..

L81: is considered to be a major contributor to oxidative stress

L89: the quality of poultry meat and processed poultry products and are a leading cause of..

L99: leading to increased water losses..

L116: evidence of the oxidative process..

L125: maintaining the health, welfare

L148: richest source..

L158: hat the natural presence…

Authors’ Response: We have now corrected minor comments raised by the Reviewer.

Reviewer 2 Report

Dear Authors,

I have read the paper entitled “Role of Dietary Methyl Sulfonyl Methane in Poultry” with interest and I believe that it could be published after revision and modifying some inaccuracies that should be corrected before acceptance of the paper. I also suggest an extensive editing of the language style because the text in the current form is very hard to read.=

General comment

-          This review article itself contains commonly available information related to free radicals, oxidative stress, low molecular and enzymatic antioxidant defense, as well as information about peroxidation and their effect on quality of poultry products. These fragments are generally well described.

-          It would be valuable to condense the information in the section 2, for instance, as “Oxidative stress as a one of the factors affecting poultry production”: and here is a place to include the sources and role of ROS in the organism, their positive and negative role, how organisms copes with their exceeded values, and how genetics (for instance towards residual feed intake), environment and pathogens induce the oxidative stress.

-          The current strategies with the use of dietary antioxidants should be directly confronted with MSM with clearly pointing out -> why MSM act as antioxidant (the mechanism), following its effectiveness in regulation the redox status observed in poultry.

-          Regarding the above, in the section entitle “4. Methyl sulfonyl methane as an antioxidant agent” the Authors mentioned (L 205-209) that sulfur-containing amino acids can neutralize ROS (MSM is a donor of S for sulfur amino acids), but there is no information about the antioxidant properties of MSM itself. Also, when MSM is postulated “as an antioxidant agent”, this should be followed by dedicated synthetic assays allowing for evaluating which mechanism of antioxidant activity is exerted (radical scavenging ability, binding of metal ions or inhibiting of peroxidation), especially as the definition of an antioxidant substance is given by the Authors (L 107). This is also important because in the animal part the Authors included the information that “The exact mechanism of MSM in attenuating the markers of oxidative stress is not well established and further exploration is needed” (L 230-237).

Other, minor comments:

¾    L 73-77: it would be worth to note that this is related to the animal's high metabolism which by itself contributes to higher levels of ROS.

¾    L 94: “malonaldehyde” -> do you mean malondialdehyde (MDA)?

¾    L 135-136: please, correct the part “dietary antioxidant antioxidants” in the sentence.

¾    L 151: please add the word “chickens organism” to the sentence.

¾    L 180: “diet-origin MSM” -> please, replace this on: “dietary MSM” (and the same in L 212).

¾    Please, verify the numeration of the selected sections, as section no. 4 is given twice (L 139 and L 229).

¾    Try to avoid phrases “As we know,” (L 414), an rephrase the sentence.

Dear Authors, I wish you successful work.

Author Response

Dear Authors,

I have read the paper entitled “Role of Dietary Methyl Sulfonyl Methane in Poultry” with interest and I believe that it could be published after revision and modifying some inaccuracies that should be corrected before acceptance of the paper. I also suggest an extensive editing of the language style because the text in the current form is very hard to read.

Authors’ Response: We have English-edited the manuscript per the Reviewer’s comment.

General comment

This review article itself contains commonly available information related to free radicals, oxidative stress, low molecular and enzymatic antioxidant defense, as well as information about peroxidation and their effect on quality of poultry products. These fragments are generally well described.

Authors’ Response: We appreciate the Reviewer’s comment.

It would be valuable to condense the information in the section 2, for instance, as “Oxidative stress as a one of the factors affecting poultry production”: and here is a place to include the sources and role of ROS in the organism, their positive and negative role, how organisms copes with their exceeded values, and how genetics (for instance towards residual feed intake), environment and pathogens induce the oxidative stress.

Authors’ Response: We agree with the Reviewer’s comments. Recently, the seminal reviews have been published as to the oxidative stress and antioxidants in general for poultry. Thus, we decided to briefly introduce the oxidative stress and antioxidant. We have now added the seminal reviews in the text and the references.

Per the Reviewer’s comment, we have now corrected as follows (Line 52).

“2. Oxidative stress as a one of the factors affecting poultry production”

The current strategies with the use of dietary antioxidants should be directly confronted with MSM with clearly pointing out -> why MSM act as antioxidant (the mechanism), following its effectiveness in regulation the redox status observed in poultry.

Authors’ Response: We appreciate the Reviewer’s comment. Dietary antioxidants have been well explored elsewhere as indicated above. Thus, we have briefly introduced dietary antioxidants, but elaborated the potential role and function of MSM in poultry as no attempts have been made to MSM in poultry. We have indicated (Figure 3) that dietary MSM improved SOD, TAC, GPX, CAT and lowered MDA in poultry. The increased or reduced level in biomarkers of oxidative stress may imply that dietary MSM has potent antioxidant activities by scavenging oxidative radicals, reducing oxidative damage, and maintaining the cell structure. It is however understood that the exact molecular mechanism of MSM in attenuating the oxidative stress is not well established. Thus, more in-depth studies on regulation the redox status in poultry by dietary MSM would help us explain or understand the molecular mechanism of MSM.

Regarding the above, in the section entitle “4. Methyl sulfonyl methane as an antioxidant agent” the Authors mentioned (L 205-209) that sulfur-containing amino acids can neutralize ROS (MSM is a donor of S for sulfur amino acids), but there is no information about the antioxidant properties of MSM itself. Also, when MSM is postulated “as an antioxidant agent”, this should be followed by dedicated synthetic assays allowing for evaluating which mechanism of antioxidant activity is exerted (radical scavenging ability, binding of metal ions or inhibiting of peroxidation), especially as the definition of an antioxidant substance is given by the Authors (L 107). This is also important because in the animal part the Authors included the information that “The exact mechanism of MSM in attenuating the markers of oxidative stress is not well established and further exploration is needed” (L 230-237).

Authors’ Response: In Section 4, we have now corrected to “4. Methyl sulfonyl methane” as this Section focuses on the general properties of MSM including free-radical scavenging antioxidant. We have now indicated two seminal reviews on in vitro antioxidant of MSM in this section.

Other, minor comments

L 73-77: it would be worth to note that this is related to the animal's high metabolism which by itself contributes to higher levels of ROS.

Authors’ Response: Birds display high metabolic rates and oxygen consumption relative to mammals, increasing ROS formation (Castiglione et al., 2020).

Castiglione, G. M., Xu, Z., Zhou, L., Duh, E. J. (2020). Adaptation of the master antioxidant response connects metabolism, lifespan and feather development pathways in birds. Nature communications, 11(1), 1-15.

L 94: “malonaldehyde” -> do you mean malondialdehyde (MDA)?

Authors’ Response: Per the Reviewer’s comment, we have now corrected (Line 90).

L 135-136: please, correct the part “dietary antioxidant antioxidants” in the sentence.

Authors’ Response: We have now corrected (Line 125-127).

“In addition, dietary antioxidants improve egg quality in laying hens and retard lipid oxidation in eggs”

L 151: please add the word “chickens organism” to the sentence.

Authors’ Response: Per the Reviewer’s comment, we have now added (Line 141).

L 180, 212: “diet-origin MSM” -> please, replace this on: “dietary MSM”.

Authors’ Response: Per the Reviewer’s comment, we have now corrected.

Please, verify the numeration of the selected sections, as section no. 4 is given twice (L 139 and L 229).

Authors’ Response: Per the Reviewer’s comment, we have now numbered all sections consecutively (Sections 5, 6, 7, 8, 9, and 10).

Try to avoid phrases “As we know,” (L 414), an rephrase the sentence.

Authors’ Response: Per the Reviewer’s comment, we have now corrected (Lines 404-405):

“It has been well established that lightness and pH in poultry meat are negatively correlated”

Round 2

Reviewer 1 Report

No comments

Reviewer 2 Report

The authors provided satisfactory responses and make many significant improvements to the manuscript.

Please consider adding “thiobarbituric acid reactive substances” in the L 232, when TBARS assay is firstly mentioned, irrespective of the explanation the principle of method described in the L 233-235.

I have no further comments.